# Antitumor, Inhibition of Metastasis and Radiosensitizing Effects of Total Nutrition Formula on Lewis Tumor-Bearing Mice

**DOI:** 10.3390/nu11081944

**Published:** 2019-08-18

**Authors:** Yu-Ming Liu, Yi-Lin Chan, Tsung-Han Wu, Tsung-Lin Li, Simon Hsia, Yi-Han Chiu, Chang-Jer Wu

**Affiliations:** 1Division of Radiation Oncology, Department of Oncology, Taipei Veterans General Hospital, Taipei 11217, Taiwan; 2School of Medicine, National Yang Ming University, Taipei 11221, Taiwan; 3Department of Life Science, Chinese Culture University, Taipei 11114, Taiwan; 4Department of Food Science and Center of Excellence for the Oceans, National Taiwan Ocean University, Keelung 20224, Taiwan; 5Division of Hemato-oncology, Department of Internal Medicine, Chang Gung Memorial Hospital, Keelung 20401, Taiwan; 6College of Medicine, Chang Gung University, Taoyuan 33320, Taiwan; 7Genomics Research Center, Academia Sinica, Taipei 11529, Taiwan; 8Taiwan Nutraceutical Association, Taipei 10596, Taiwan; 9Department of Nursing, St. Mary’s Junior College of Medicine, Nursing and Management, Yilan 26647, Taiwan; 10Institute of Long-Term Care, Mackay Medical College, New Taipei City 25245, Taiwan; 11Department of Medical Research, China Medical University Hospital, China Medical University, Taichung 40402, Taiwan; 12Department of Health and Nutrition Biotechnology, Asia University, Taichung 41354, Taiwan; 13Graduate Institute of Medicine, Kaohsiung Medical University, Kaohsiung 80708, Taiwan

**Keywords:** non-small-cell lung cancer (NSCLC), radiotherapy, radiosensitization, total nutrition formula (TNuF), cachexia

## Abstract

Non-small-cell lung cancer (NSCLC) causes high mortality. Radiotherapy is an induction regimen generally applied to patients with NSCLC. In view of therapeutic efficacy, the outcome is not appealing in addition to bringing about unwanted side effects. Total nutrition is a new trend in cancer therapy, which benefits cancer patients under radiotherapy. Male C57BL/6JNarl mice were experimentally divided into five groups: one control group, one T group (borne with Lewis lung carcinoma but no treatment), and three Lewis lung carcinoma-bearing groups administrated with a total nutrition formula (T + TNuF group), a local radiotherapy plus daily 3 Gy in three fractions (T + R group), or a combination TNuF and radiotherapy (T + R + TNuF group). These mice were assessed for their mean tumor volumes, cachectic symptoms and tumor metastasis. TNuF administration significantly suppressed tumor growth and activated apoptotic cell death in NSCLC-bearing mice under radiation. The body-weight gain was increased, while the radiation-induced cachexia was alleviated. Analysis of mechanisms suggests that TNuF downregulates EGFR and VEGF signaling pathways, inhibiting angiogenesis and metastasis. In light of radiation-induced tumor cell death, mitigation of radiation-induced cachexia and inhibition of tumor cell distant metastasis, the combination of TNuF and radiotherapy synergistically downregulates EGFR and VEGF signaling in NSCLC-bearing mice.

## 1. Introduction

Lung cancer is the most prevalent cancer globally [1], in which non-small cell lung cancer (NSCLC) accounts for 85% of total lung cancer [2]. Because of no effective early detection, two thirds of total lung cancers diagnosed with NSCLC at stages IIIB or IV. The opportunity for curative surgery has considerably dwindled by this stage [3], making the 5-year survival rate as low as 10–15% [4]. Moreover, NSCLC has a propensity to metastasize to distal organs, and no effective therapies are able to contain the disease at this stage [5]

Radiotherapy is an alternative therapy for inoperable patients with stage I NSCLC, or for NSCLC patients with locally advanced unresectable lesions, or poor prognostic factors [6,7,8]. After irradiation with high-energy photon beams, DNA double-strand breaks (DSBs) generally result, of which the extent of DSBs dictates cell survival [9]. Despite the undeniable merits of radiotherapy, the serious side effects experienced by patients, such as the negative impact on the immune system and bone marrow suppression, as well as harm to the gastrointestinal tract, make this therapy difficult to tolerate and complete [10]. The treatment is further compromised because of poor radiosensitivity of cancer cells [11,12,13,14,15]. Normal cells/tissues are actually more sensitive to radiation than tumor cells/tissues, thus resulting in treatment failure [16]. What makes the matter worse is that NSCLC has long been regarded as a radioresistant malignancy [6,17], making radiotherapy a suboptimal treatment [18]. Nevertheless, if a treatment can sensitize tumor cells, but not normal cells, to radiation, the treatment could enhance the position of radiotherapy for the treatment of cancer patients.

Naturally edible radiosensitizers are considered safer than chemically synthesized counterparts. Natural products that possess antioxidant or immune-enhancing effects, are potential candidates, for example, ω-3 polyunsaturated fatty acids, soy isoflavones and curcumin. ω-3 Polyunsaturated fatty acids have been shown to enhance the cytotoxic effect of radiotherapy in both radio-sensitive and radio-resistant colorectal cancer cells. The effect was attributed to lipid peroxidation by modulating inflammatory responses and apoptosis [19]. The effect of soy isoflavones was correlated to inhibition of DNA repair by inhibiting APE1/Ref-1 and HIF-1α survival responses in cancer cells [20,21]. Curcumin is also known to downregulate LIG4/PNKP and upregulate XRCC5/CCNH in irradiation-treated-colon cancer, leading to tumor-growth inhibition and apoptosis [22]. 

Our previous studies showed that patients with malnutrition not only had a reduced tolerance to radiation, but also demonstrated increased irradiation-induced morbidities such as mucositis, impaired swallowing function, declined eating ability, xerostomia, dysgeusia and nausea, which resulted in poor survival in NSCLC, and in patients with head and neck cancers and nasopharyngeal carcinoma [23,24,25]. Micronutrient deficiency is also related to cancer cachexia and tumor progress. For example, the level of selenium or coenzyme Q_10_ (CoQ_10_) is inversely proportional to the level of cachexic cytokines and the prevalence rate of metastases. In contrast, adequate supplementation of selenium and/or CoQ_10_ reverses the trend, thereby facilitating inhibition of cachexia progression and metastatic diffusion [26,27]. 

Total nutrition formula (TNuF) is a diet fortified with protein, energy and micronutrients (vit A, selenium, and CoQ_10_). The effect of TNuF on NSCLC tumor cells in mice under radiotherapy was used to index the extent of tumor progress, cancer cachexia and distant metastasis. Our results suggest that TNuF is a potent radiosensitizer with a marked anti-tumor/cachexia/metastasis effect on NSCLC-bearing mice. 

## 2. Materials and Methods

### 2.1. Cells and Cell Culture

Lewis lung carcinoma (LLC, ATCC CRL-1642) cells, which is a cell linage derived from a C57BL/6 strain mice lung cancer, were obtained from the Bioresource Collection and Research Center (BCRC, Hsinchu, Taiwan). LLC cells were maintained following the BCRC instructions. In brief, LLC cells were cultured in Dulbecco’s modified Eagle’s medium (DMEM) supplemented with 10% fetal bovine serum at 37 °C in a humidified incubator containing 5% CO_2_. 

### 2.2. Mice and Tumor Model

Male C57BL/6JNarl mice (at 5–6 weeks old) were ordered from the National Laboratory Animal Center (NLAC, Taipei, Taiwan, ROC) and housed in a climate-controlled room (12:12 dark–light cycles with a constant room temperature of 21 ± 1 °C). Mice adapted themselves to a new environment and diet at least 4 days before experiments were started. Mice were fed with food and water *ad libitum*. All animal experimental protocols were reviewed and approved by the Institutional Animal Care and Use Committee (IACUC) of National Taiwan Ocean University (NTOU), and the study conformed to the guidelines of the protocol IACUC-105020 approved by the IACUC ethics committee of NTOU. To assess the effect of radiotherapy with/without TNuF on tumor growth, mice were divided into five weight-matched groups (*n* = 6 per group) (Figure 1): (1)C group: control receiving normal saline;(2)T group: LLC-inoculated mice received normal saline;(3)T + TNuF group: LLC-inoculated mice treated with 1 g TNuF administration orally once daily;(4)T + R group: LLC-inoculated mice treated with a conventional fractionated dose of 3.3 Gy, one fraction per day, three days per week group;(5)T + R + TNuF group: LLC-inoculated mice treated with the combination of TNuF and radiotherapy.

To minimize experimental error, 5 × 10^5^ LLC cells in 100 μL were unilaterally inoculated into the right posterior flank region of each BALB/c nude mouse. At the 7th day post tumor-cell inoculation, mice were given 1 g/mice/day TNuF in 200 μL normal saline via oral gavage until sacrifice. Fourteen days later (on the 21th day), the mice were anesthetized and sacrificed, where the organs were removed, weighed, and then stored at 20 °C for further analysis. The lungs were subjected to histological analysis to assess lung metastasis. 

### 2.3. Components of the TNuF

The TNuF supplementations were packaged, supplied, and shipped by New Health Enterprise Inc. (Irvine, CA, USA). TNuF is a protein- and energy-dense oral nutritional supplement that contains several ingredients, including Vitamin A, selenium, and CoQ_10_. Table 1 shows the compositions of the supplement. 

### 2.4. Delivery of Radiation

The animals were treated with different protocols. A local radiotherapy with daily fractions of 3 Gy in three fractions were given on the 8th, 10th, and 12th days. The radiation field encompassed the gross tumor plus a 0.3- to 0.5-cm margin over right posterior flank region of mice. Six MeV of electron-beam energy was delivered by a linear accelerator (Clinac 1800; Varian Associates, Inc., Palo Alto, CA, USA). The dose rate was 2.4 Gy/min. Full electron equilibrium was ensured for each fraction using a parallel-plate ionization chamber (PR-60C; Capintel, Inc., Ramsey, NJ, USA). 

### 2.5. Assessment of Tumor Volume and Metastasis

The tumor volume [V = L (longest diameter) × W^2^ (shortest diameter) × 0.5] was measured once every two days. Following sacrifice, the tumors were excised and weighed. The lungs were harvested and the surface nodules were counted to evaluate the metastatic spread of the tumor. In brief, whole tumor and tumor-bearing lungs were manually inflated with and fixed in 4% paraformaldehyde and embedded. Paraffin-embedded lungs were serially sectioned at 350 mm and histologically examined with hematoxylin and eosin. Microscopic observations were carried out at 200 × magnifications. 

### 2.6. Blood Sample Preparation and Analyses

Blood samples (0.5 mL for each) were collected and counted using a blood cell analyzer (Symex K-1000, Sysmex American, Mundelein, IL, USA), by which the measurements included the counts of white blood cells (WBCs), lymphocytes, monocytes and neutrophils. All assays were performed following the protocols provided by the manufacturer. 

### 2.7. Immunofluorescence Assay

Sections were prepared and blocked by using a blocking buffer solution for 1 h at room temperature and then incubated with a staining buffer solution containing primary antibody at 1:200 dilution for 24 h. The extra primary antibody was washed away by PBS. Sections were stained with a staining buffer solution containing secondary antibody at 1:100 dilution for 24 h at room temperature and then washed with PBS. The primary antibodies that were used are as follows: rabbit anti-mouse EGRF (Santa Cruz, Santa Cruz, CA, USA), rabbit anti-mouse VEGF (Santa Cruz) rabbit anti-mouse CD 31 (Santa Cruz) and rabbit anti-mouse HIF-1α (Santa Cruz). The secondary antibodies were FITC-conjugates goat anti-rabbit IgG (Sigma, Saint Louis, MO, USA). 

### 2.8. RNA Extraction and Real-Time PCR

RNAs were extracted from total tumors or liver tissues with the Rneasy mini kit (Qiagen, Germantown, MD, USA), on which cDNA was synthesized using the M-MLV reverse transcriptase (Promega, Madison, WI, USA) and oligo-dT15-primers (Promega, Madison, WI, USA). Real-time PCR was performed using the Bio-Rad iCycler iQ system. Quantitative real-time PCR analysis was carried out in a 25-μL reaction consisting of 12.5 μL iQ SYBR Green Supermix (Bio-Rad), 5 μL cDNA, RNase-free water, and 100 μM of each primer. Values were normalized to GAPDH mRNA amount. The oligonucleotide primers for mouse Bax (5’-ACCAAGAAGCTGAGCGAGTGT-3’ and 5’-ACAAAGATGGTCAC GGTCTGC C-3’), mouse Bcl-2 (5’-CCTCACCAGCCTCCTCAC-3’ and 5’-ACTACCTGCGTTCTCCTCTC-3’), mouse VEGF (5’-GATGTATCTCTCGCTCTCT C-3’ and 5’-CTTCTCAGGACAAGCTAGTG-3’), mouse caspase-3 (5’-GGAGATGGC TTGCCAGAAGA- 3’ and 5’-ATTCCGTTGCCACCTTCCT-3’), and mouse GAPDH (5’-ACAATGAATACGGCTACAG-3’ and 5’-GGTCCAGGGTTTCTTACT-3’) were used according to previously published sequences. 

### 2.9. Statistical Analysis

All experiments were performed three times, each time in triplicate. Data were analyzed by multivariate ANOVA test. If a significant difference was found, the least significant differences (LSD) multiple comparison test was used to identify significant groups. Statistical analyses were performed using The Statistical Software Package for the Social Sciences, version 12.0.1 for Windows (SPSS Inc., Chicago, IL, USA). A *p*-value <0.05 is considered statistically significant. 

## 3. Results

### 3.1. Effects of TNuF in Combination with Radiotherapy on Spontaneous Apoptosis and Tumor Growth

Primary tumor formation was detected 7 days after tumor inoculation in all mice. The tumor-bearing mice were grouped into three, which were individually treated with TNuF (T + TNuF), radiotherapy (T + R), or TNuF plus radiotherapy (T + R + TNuF) (Figure 2A). 

The mean tumor volume reached 1 cm^3^ in mice of T group at 14 days after tumor injection (Day 21). In contrast, both tumor growth and tumor weight were reduced in T + R and T + TNuF groups, when compared to those in T group. The mean tumor volume was reduced to 600 mm^3^ in TR group at the 21th day, while the growth of tumors had no significant difference (when compared to that of T group, *p* > 0.01; Figure 2A). The growth of LLC cells was suppressed after TNuF administration. The suppression of tumor growth in T + R + TNuF was more significant than that in other groups. Additionally, the tumor weight in T group was 0.74 ± 0.05 g, whereas that in T + TNuF, T + R or T + R + TNuF group was reduced by 50.43%, 46.83% or 60.63%, respectively, when compared with that in T group (Figure 2B). The reduction was most significant in T + R + TNuF, suggesting co-treatment of TNuF and irradiation has a synergistic antitumor effect. 

The expressions of Bax, Bcl-2 and caspase 3 in tumor cells were gauged using QPCR. The level of Bcl-2 in tumor cells was influenced with the treatment of TNuF, irradiation or both to various extents. The level of Bax in tumor cells was increased with the treatment of TNuF or TNuF plus irradiation, while there was no change with irradiation alone. QPCR analysis showed that caspase 3 mRNA was slightly elevated in tumors treated with TNuF plus irradiation, despite no statistic difference compared to that in control tumors (Figure 2C–E). Added together, the antitumor activity of TNuF and/or irradiation can be attributed to the apoptosis-promoting effect, of which TNuF enhances both radiosensitivity and radiation-induced apoptosis.

### 3.2. Cachectic Symptoms and Hematology Parameters

The changes of body weights among testing animals are summarized in Figure 3A. The body weights of control, T, T + TNuF, T + R and T + R + TNuF mice were increased by 5.50%, 6.30%, 2.15%, −1.96% and 0.15%, respectively, at the due course. In comparison with T group, the body weights of mice treated with T + TNuF or T + R + TNuF were significantly increased, substantiating the effect of TNuF on inhibition of tumor growth (Figure 3B). The loss of body weights in LLC tumors-bearing mice may be due to mass loss of gastrocnemius medialis (GM) muscles (Figure 3A,C). Although the reduction of tumor load with the irradiation treatment was as high as 46.83% (Figure 3B), the body weight and muscle mass were not increased. The extent of losses of body weights and muscle masses in mice receiving irradiation (T + R group) was comparable to that of tumor bearing mice (T group). In contrast, the body weights of mice receiving T + TNuF or T + R + TNuF were increased, respectively, by 1.03 ± 0.08% or 4.24 ± 0.08% (Figure 3B), of which the GM weights were increased by 26% or 25% (Figure 3C).

As shown in Table 2, the levels of TNF-α and IL-6 in T mice, which are cachectic and hematologic indicators, were higher than those in mice treated with TNuF, irradiation, or the combination. In contrast, albumin in T mice was significantly lower than that in mice treated with TNuF or the combination, concluding that TNuF effectively alleviates cancer-associated cachexia. 

Hematologic parameters, RBC, WBC and neutrophil-lymphocyte count ratio (NLR), were used to index the extent of cachexia [28,29]. As shown in Table 2, the total RBC counts in all groups were similar (*p* < 0.05), suggesting that both the tumor inoculation and treatments did not cause a significant difference in the number of RBC. The WCB counts were 3.64 ± 1.27, 5.11 ± 0.60, 4.42 ± 0.76 and 5.87 ± 0.64 × 10^3^/mm^3^ for T, T + TNuF, T + R and T + R + TNuF treated mice, respectively, which in general were lower than that of healthy control. However, leukopenia was decreased in the group treated with TNuF, suggesting that TNuF alleviates cachectic leukopenia in tumor-bearing mice. 

NLR is a biomarker that is used to gauge prognosis [28], of which the level is negatively correlated to the survival rate of inoperable cancers [29]. The levels of NLR in T + TNuF, T + R and T + R + TNuF group (0.93 ± 0.00, 0.95 ± 0.02, and 0.99 ± 0.01, respectively) were statistically lower than that in T group, suggesting that TNuF and irradiation counteract negative influences of NLR (Table 2). 

### 3.3. Inhibition of Growth and Angiogenesis of C57BL/6JNarl Mice by TNuF 

To understand how TNuF or the combined treatment negatively regulates neoplastic processes and angiogenesis, we set out to examine possible pathways that regulate productions of epidermal growth factor receptor (EGFR), vascular endothelial growth factor (VEGF), HIF-1α and CD31 in tumor tissues. The expressions of EGFR, VEGF, HIF-1α and CD31 were significantly decreased in the groups treated with TNuF, irradiation, or both when compared to those in T mice (Figure 4A,B), although there were no significant differences amid three groups. In addition, the mRNA levels of VEGF in the groups treated with TNuF, radiation, or the combination were normalized to be 83%, 68%, or 90% relative to that of T group (Figure 4C), suggesting that the treatments are closely associated with inhibition of EGFR and its downstream modulation on HIF-1α/VEGF/CD31. 

### 3.4. Inhibition of Lung Metastatic Colonization of LLC Cells in C57BL/6JNarl Mice by TNuF

We further examined what effects would result on the lung metastatic colonization of LLC cells when TNuF, irradiation, or both were applied. As shown in Figure 5A, the lung weights of the control and T groups are 0.11 ± 0.00 and 1.17 ± 0.01 g, respectively, where the latter is higher than the former (*p* < 0.001). Tumors, in fact, grew considerably fast, while the growth rate was significantly retarded by 17.77%, 23.70% or 42.44%, when TNuF, irradiation or the combination was supplied, respectively (*p* < 0.001). The increased lung mass was proportional to the increased lung tumor or tumor nodule mass, where the mean number of nodules in lungs treated by T, T + TNuF, T + R or T + R + TNuF is 2.83 (range 0–8), 2.17 (range 0–4), 0.83 (range 0–1) or 0.67 (range 0–2), respectively (Figure 5B,C). The number of metastatic foci was correlated to the treatment, and the lung weight was positively proportional to the number of tumor nodules in lung (Figure 5D, r = 0. 693, *p* < 0.001). As shown in Figure 5D, the H&E staining was consistent with above findings; namely, the tumor load was decreased in the mice treated with TNuF, radiation or the combination. 

In immunofluorescence assay, the fluorescence intensities of EGFR, VEGF, CD31 and HIF-1α in lungs, likewise, were decreased in mice treated with TNuF, irradiation or the combination (Figure 5E,F). While the intensity of EGRF in T + R mice is higher than that in control, all the animals treated with TNuF or T + R + TNuF show attenuated neoplastic processes and angiogenesis as opposed to T mice (Figure 5E,F). Taken together, we conclude that TNuF plays a significant role in regulating EGFR and its subsequent modulation on HIF-1α/VEGF/CD31 in lung metastases. 

## 4. Discussion

While radiotherapy is a standard regimen for patients with NSCLC, it often loses its battle to NSCLC because of radioresistance [30,31]. The antitumor effect of irradiation or TNuF can be depicted in the tumor growth curve, where the tumor volume of T mice increases with time as opposed to that of T + R mice that increases sluggishly in the first 17 days but rapidly increases in the last few days, consistent with the general outcome of radiotherapy [32]. The patterns for both T + TNuF and T + R + TNuF mice, however, are different, and the tumor grows at a much slower rate than that observed in groups not receiving TNuF (Figure 2). In contrast to mice in group T, the growth of tumors for groups subject to irradiation or TNuF significantly reduced, and the irradiation efficacy in LLC-bearing mice was simultaneously improved. TNuF in conjunction with irradiation showed an additive effect, whereby the expression of pro-apoptotic Bax and caspase 3 genes increased, as opposed to anti-apoptotic Bcl-2 that was decreased. These factors significantly counteract negative influences of LLC tumor cells. Because irradiation alone is less efficient, the TNuF-modulated apoptosis is an ideal complement to radiotherapy. An appropriate radiosensitizer is considered as an agent that is able to balance both radiosensitization and radiotherapeutic toxicity [33,34,35,36,37]. TNuF that not only inhibits tumor proliferation but also improves irradiation-associated cachexia appears to be an ideal radiation sensitizer in its own right. 

More than half of cancer deaths are attributable to cancer-associated cachexia. Though radiotherapy is an effective treatment, it simultaneously induces many unwanted sequelae, such as involuntary loss of body weight, loss of homeostatic control of both energy and protein balance, thus leading to progressive functional impairment, cancer-related mortality, treatment-related complications and poor quality of life [38,39]. T + R mice that experienced more significant body-weight loss than T mice were consistent with the above notion. Weight loss is a consequence of distress in cancer patients, while inadequate nutrient intake is the main cause of weight loss. van Dijk et al. reported that reduction of dietary nutrient intake (e.g., vitamin A, folate, selenium, etc.) has a negative impact on force-generating capacity, fatigue resistance, physical activity and muscle mass [40]. Given TNuF that is rich in protein, vitamins A, selenium and coenzyme Q_10_, the oral administration of TNuF considerably prevents cancer/irradiation-induced cachexia. The rationale can be correlated to decreases of both skeletal muscle depletion and pro-inflammatory mediators (TNF-α and IL-6), and homeostasis of the ratio of NLR. 

EGFR is overexpressed in NSCLC cells by 40–80% [41], underscoring the potency of EGFR inhibitors in cancer cell proliferation [42]. Signaling of VEGF is positively correlated to expression of EGFR; HIF-1α, on the other hand, is a transcription factor that induces VEGF in hypoxic conditions and thus promotes angiogenesis [43]. The increased level of EGFR and vascular related proteins would stimulate tumor cell proliferation and angiogenesis, thereby facilitating tumor cell growth and metastasis. Both vascular targeting drugs and EGFR inhibitors were designed to enhance sensitivity of tumor cells upon radiation. TNuF appears to be vascular targeting compounds and/or EGFR inhibitors in this study. Our results conclude that the combination of TNuF and radiotherapy synergistically inhibits EGFR and VEGF signaling. Despite that a handful of reports has suggested that radiation sensitivity of tumor cells is negatively correlated to the expression levels of EGFR and VEGF [44,45], our study, nevertheless, firstly report that oral administration of TNuF effectively suppresses EGFR upon radiation. 

Given that VEGF is a potent angiogenic factor promoting tumor expansion and secondary malignancy, an effective anti-VEGF agent should inhibit angiogenesis-dependent tumor growth and metastasis. It has been reported that the radiation-activated PI3K/Akt/mTOR signaling pathway upregulates VEGF-C overexpression in lung cancer cells promoting endothelial cell proliferation [46]. In contrast, Xiang et al. demonstrated that the radiation downregulates the expressions of HIF-1α and VEGF in tumors thus inhibiting lung tumor angiogenesis [47]. Although the conclusion is inconsistent in literatures, our results, however, agree that the radiation considerably suppresses the VEGF expression at the both gene and protein levels (Figure 4). On the other hand, TNuF exhibits a VEGF inhibiting effect on tumor tissues, whereas the synergistic effect is not significant when TNuF is combined with radiation. TNuF was experimentally shown to suppress metastasis of tumor cells and expression of tumor-induced VEGF as well as expression of EGFR, HIF-1α and CD31 in lung tissues (Figure 5). The overall outcome is that TNuF constrains cancer cells from invasion and metastasis by suppressing epithelial cell proliferation and blood vessel formation despite that the major active component and the precise anti-metastasis mechanism of TNuF remain unclear. Nonetheless, recent studies have shown that selenium plays an important role in preventing cancer from metastasis. It has also been reported that dietary supplementation with high-selenium soy protein resulted in 45% reduction of lung metastases of melanoma cells in mice [48]. Dietary supplementation with selenium in the form of methylseleninic acid (MSeA) similarly showed a reduction of lung metastasis by 70% in an MMTV-PyMT mice model experiment [49]. Moreover, doxorubicin-loaded selenium nanoparticles were shown able to inhibit breast tumor metastasis via down-regulation of VEGF-VEGFR2 in nude mice [50]. We consider that the anti-metastasis potential of selenium in TNuF is deserved to undergo in-depth evaluation in future studies.

## 5. Conclusions

In conclusion, our current results suggest that TNuF is an ideal radiosensitizer, as it can inhibit cancer invasion and metastasis by targeting EGFR and VEGF signaling. The TNuF regimen significantly improves cachectic symptoms, including maintenance of body weight, increase of serum albumin, decrease of inflammatory cytokines and rescue of cachectic leukopenia. Importantly, TNuF inhibits colonization of lung metastasis by reducing EGFR and VEGF/CD31/HIF-1α expressions (Figure 6). Thus, TNuF is a radiosensitizer and radioprotector, which in conjunction with radiotherapy should maximize treatment outcomes.

## Figures and Tables

**Figure 1 nutrients-11-01944-f001:**
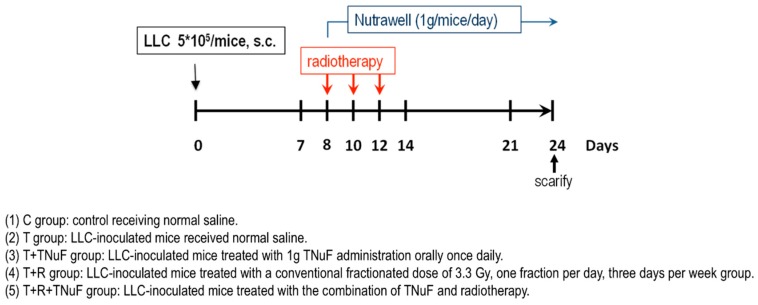
The treatment schedule of total nutrition formula (TNuF) and irradiation in tumor-bearing mice.

**Figure 2 nutrients-11-01944-f002:**
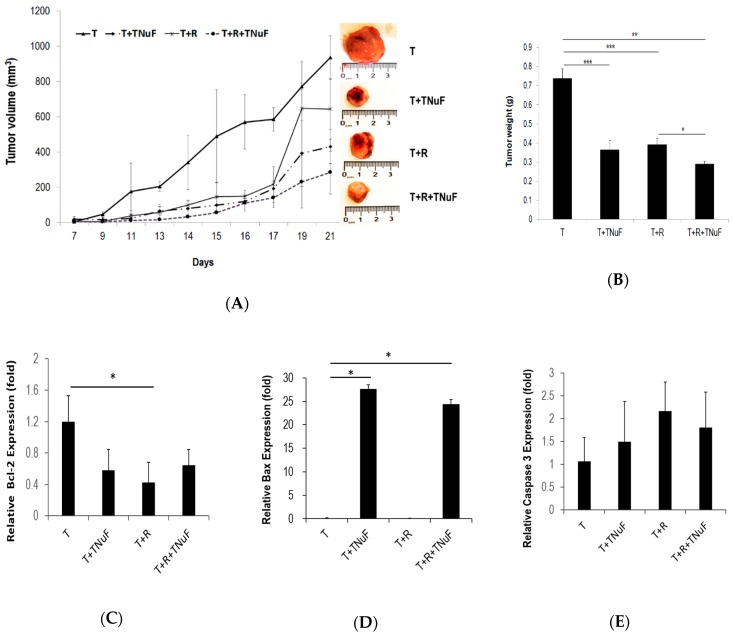
The effects of TNuF and/or irradiation on the growth of xenograft tumor. (**A**) Tumor volume growth curve with various treatments and representative tumors from the mice in due course. (**B**) The tumor weights measured at the end point. mRNA levels of (**C**) Bcl-2, (**D**) Bax and (**E**) caspase 3 in the tumor tissues of mice treated with TNuF, irradiation, and the combination of TNuF and irradiation. Data are expressed as the means ± SD. * *p* < 0.05, ** *p* < 0.01, *** *p* < 0.001 when compared to the indicated group.

**Figure 3 nutrients-11-01944-f003:**
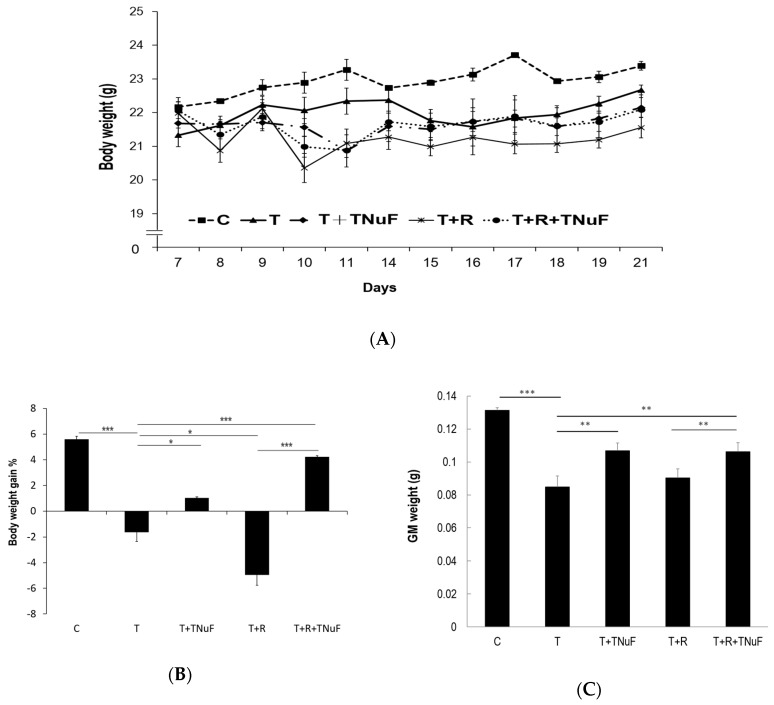
The cachectic symptoms after treatment of TNuF and/or irradiation in Lewis lung carcinoma (LLC) bearing mice. (**A**) The body weights of tumor-bearing nude mice with the treatments during the course. (**B**) Body-weight gain and (**C**) gastrocnemius medialis (GM) muscles mass for the mice in due course. Data were derived from three independent experiments and presented by mean ± SEM. * *p* < 0.05, ** *p* < 0.01, *** *p* < 0.001 when compared to the indicated group.

**Figure 4 nutrients-11-01944-f004:**
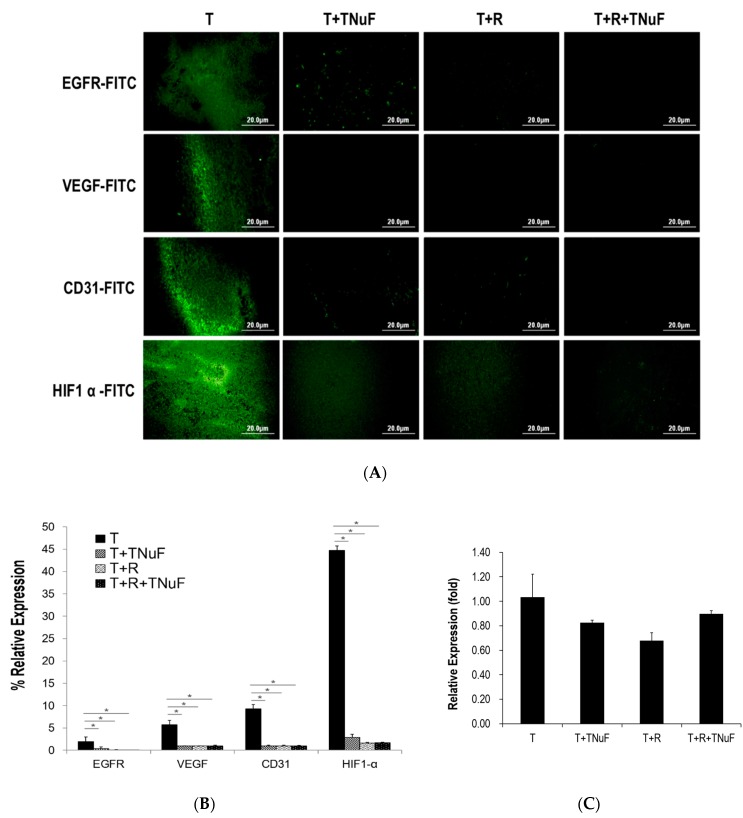
Expression of neoplastic processes and angiogenesis proteins in tumor tissues of tumor bearing mice treated with TNuF and/or plus irradiation. (**A**) Immunofluorescence analysis for tumor tissues treated with TNuF and/or plus irradiation. Images are shown at ×200 magnification. (**B**) Protein expression levels that are quantified and expressed as a relative change to the LLC cell-inoculated group. Asterisk (*) indicates a significant difference (*p* value < 0.05) when compared to the control group. (**C**) Quantification of mRNA levels of vascular endothelial growth factor (VEGF) by qRT–PCR in the tumor tissues of all mice treated with TNuF, irradiation, and the combination of TNuF and irradiation.

**Figure 5 nutrients-11-01944-f005:**
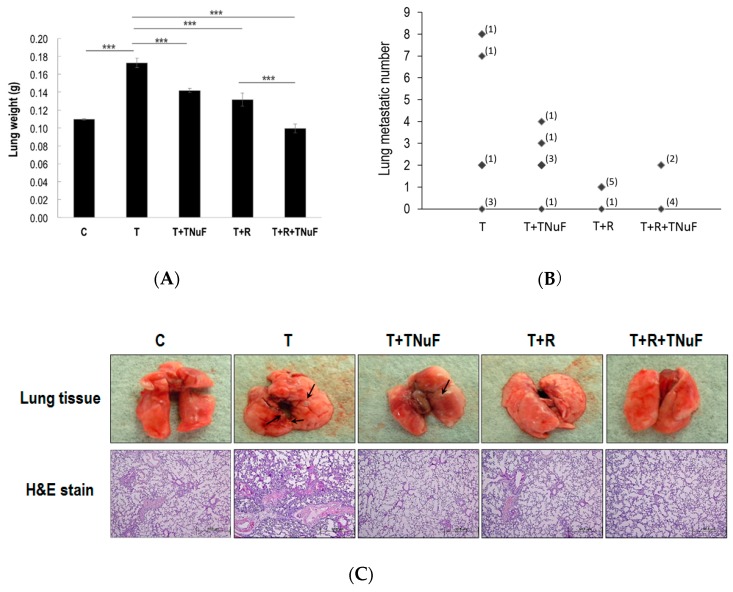
TNuF and/or plus irradiation affected lung metastatic colonization of LLC cells in C57BL/6JNarl mice. (**A**) Lung weight from the mice of each group at the end time-point. (**B**) Quantification of metastatic nodules present in the bilateral lungs of mice; each point represents an individual animal. (**C**) Lung appearance (up) and histology (H + E stain; down) in LLC inoculated C57BL/6JNarl mice. The solid tumors (indicated by arrows) were spotted on multiple sites in mice. Five representative samples are shown. (**D**) Correlation between lung weight and tumor nodule number in lungs. (**E**) Immunofluorescence analysis for lung tissues treated with TNuF and/or plus irradiation. Images are shown at ×200 magnification; (**F**) Protein expression levels that are quantified and expressed as a relative change to the LLC cell-inoculated group. Data were derived from three independent experiments and presented by mean ± SEM. * *p* < 0.05, *** *p* < 0.001 when compared to the compared group.

**Figure 6 nutrients-11-01944-f006:**
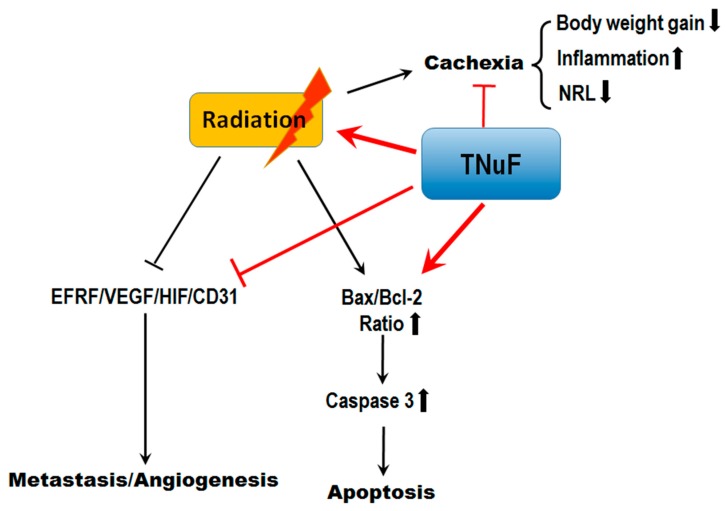
Proposed antitumor, metastasis inhibitory and cachexia improving model of TNuF in non-small-cell lung cancer (NSCLC)-bearing mice under radiation. In NSCLC-bearing mice, the TNuF administration significantly suppressed tumor growth and activated apoptotic cell death in NSCLC-bearing mice under radiation. The body-weight gain, inflammation and NRL were improved after TNuF administration, while the radiation-induced cachexia was mitigated. Mechanism analysis suggests that TNuF downregulates epidermal growth factor receptor (EGFR) and VEGF signaling pathways, inhibiting angiogenesis and metastasis. Accordingly, the present study may provide information regarding the TNuF administration with radiation therapy in lung cancer, and the regulation of EGFR/VEGF may be a promising strategy for treating lung cancer metastasis.

**Table 1 nutrients-11-01944-t001:** Major components of the trial total nutritional formula (Nutrawell) supplement.

NutraWell Powder	Serving Size: 5 Scoops (75 g)
Components (units)	Amount Per Serving
Total Calories (Kcal)	298
Total Fat (g)	8.7
Cholesterol (mg)	0
Sodium (mg)	350
Total Carbohydrate (g)	38
Protein (g)	17
Vitamin A	
Retinyl Acetate (IU)	1167
β-carotene (IU)	1000
Vitamin C (mg)	100
Vitamin D (IU)	150
Vitamin E (mg)	10
Vitamin K (mcg)	30
Vitamin B1 (mg)	1.2
Riboflavin (mg)	1.2
Niacin (mg)	12
Vitamin B6 (mg)	1.3
Folate (mcg)	100
Vitamin B12 (mcg)	1.5
Biotin (mcg)	50
Pantothenic Acid (mg)	3
Choline (mg)	250
Calcium (mg)	350
Iron (mg)	4
Phosphorous (mg)	200
Iodine (mcg)	40
Magnesium (mg)	100
Zinc (mg)	6
Selenium (mcg)	65
Copper (mcg)	250
Manganese (mg)	1.5
Chromium (mcg)	90
Molybdenum (mcg)	56.3
Potassium (mg)	550
Coenzyme Q10 (mg)	20

**Table 2 nutrients-11-01944-t002:** Effects of TNuF and irradiation on cachectic and hematological parameters in the LLC xenografted mouse model.

Treatment	Cachectic Parameters			Hematology Parameter			
	Albumin (g/dL)	TNF-α (pg/mL)	IL-6 (pg/mL)	RBC (10^12^/L)	WBC (10^3^/mm^3^)	Lym (%)	NLR
Con	2.88 ± 0.07	0.00 ± 0.00	0.00 ± 0.00	7.39 ± 1.63	9.20 ± 0.32	82.10 ± 0.99	0.24 ± 0.01
T	2.57 ± 0.06 ^a^	2.50 ± 1.31 ^a^	20.05 ± 8.13 ^a^	7.52 ± 1.27	3.64 ± 1.27 ^a^	63.24 ± 1.67 ^a^	1.17 ± 0.05 ^a^
T + TNuF	3.18 ± 0.05	0.63 ± 0.49 ^b^	15.87 ± 9.09 ^b^	7.18 ± 1.28	5.11 ± 0.60	73.26 ± 0.85 ^b^	0.93 ± 0.00 ^b^
T + R	2.75 ± 0.06 ^b^	1.42 ± 0.73 ^b^	6.16 ± 1.23 ^b^	7.20 ± 1.02	4.42 ± 0.76	67.72 ± 0.61 ^b^	0.95 ± 0.02 ^b^
T + R + TNuF	2.85 ± 0.06	0.40 ± 0.17 ^b^	10.87 ± 4.17 ^b^	8.19 ± 0.66	5.87 ± 0.64	75.21 ± 0.11 ^b^	0.99 ± 0.01 ^b^

TNF, tumor necrosis factor; IL, interleukin; RBC, red blood cell count; WBC, while blood cell count; NLR, neutrophil-lymphocyte count ratio. Data are expressed as means ± S.E. 6 mice were used per treatment (*n* = 6). ^a^
*p* < 0.05 versus the control group. ^b^
*p* < 0.05 versus the T group.

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
