# Peer review of "Antitumor, Inhibition of Metastasis and Radiosensitizing Effects of Total Nutrition Formula on Lewis Tumor-Bearing Mice"

_nutrients, 2019, doi:10.3390/nu11081944_

Round 1

Reviewer 1 Report

This is an animal study to examine the effect of TNuF. Major concerns include:

1.     Sample numbers need to be discussed. How many mice per each group?

2.     Figure 2 shows the expression levels of different apoptosis markers. The connections between different treatments and apoptosis need to be discussed.

3.     What is the red blood cells counts in different groups? Did the authors measure other hematology parameters besides what has been shown in Table 2?

4.     Figure 4 and Figure 5 show the immunofluorescence images of several marker proteins. These results are important to this study. To quantify the protein levels, it is better to use Western Blot analysis.

5.     Font styles and sizes are different between figures. Figures need to be carefully edited to make them have uniform style and format.

6.     This manuscript does not read well and needs to be carefully edited before resubmission.

Author Response

Comment 1: Sample numbers need to be discussed. How many mice per each group?

Reply: We thank the reviewer’s suggestion for this suggestion. There are six mice in each group, which has been added in line 20-21, page 5, in this revised manuscript.

Comment 2: Figure 2 shows the expression levels of different apoptosis markers. The connections between different treatments and apoptosis need to be discussed.

Reply: Again, we thank the reviewer’s suggestion with respect to Figure 2, which has been addressed in discussion section from line 7-17 page 19 in this revised version. In general, we firstly showed expression levels of different apoptotic markers in tumor cells using QPCR (Figure 2C-E). The results demonstrated that the antitumor activity of TNuF and/or irradiation is attributable to its apoptosis-promoting effect, in which TNuF enhances both radiosensitivity and radiation-induced apoptosis.

Comment 3: What is the red blood cells counts in different groups? Did the authors measure other hematology parameters besides what has been shown in Table 2?

Reply: We thank the reviewer for bringing this point to our attention. The result with respect to red blood cell counts has been described in lines 1-3 page 14 and in Table 2 in this revised manuscript. We have indeed measured other hematology parameters, such as HGB, HCT, MCV, MCH, MCHC, RDW%, PLT, MPV, etc., while there are no differences between each experimental group. Therefore, we didn’t mention the data in the manuscript.

Comment 4: Figure 4 and Figure 5 show the immunofluorescence images of several marker proteins. These results are important to this study. To quantify the protein levels, it is better to use Western Blot analysis.

Reply: We appreciate the reviewer’s comment. We understand that western blotting is one of standard methods to quantify protein expression. The fact is that western blotting had been performed using a routine sample setup composed of protein lysates from a limited number of tumor tissues and cell lines. However, the outcomes were nothing better than that done by immunofluorescence (IF) in the scopes of quantified localization, relative expression, and activation states of target proteins. IF uses antibodies and fluorescent detection to index the localization, relative expression, and activation states of target proteins in fixed cells or tissues. IF is also used to visualize target proteins and other biomolecules within tumor tissue samples. Given that IF is routinely used to present graphical and quantifiable data, we think that the IF results presented herein are appropriate.

Comment 5: Font styles and sizes are different between figures. Figures need to be carefully edited to make them have uniform style and format.

Reply: We thank the reviewer’s suggestion. The figures in question have been reformatted to have consistent font style and size.

Comment 6: This manuscript does not read well and needs to be carefully edited before resubmission.

Reply: We thank the reviewer’s advice. The language has been rigorously revised in this revised version.

Reviewer 2 Report

In the manuscript entitled Antitumor, metastasis inhibitory and radiosensitizing effects of total nutrition formula on Lewis tumor-bearing mice the authors have found that

 total nutrition formula (TNuF)  administration significantly suppressed tumor growth and activated apoptotic cell death in NSCLC-bearing mice under radiation. The body-weight gain was increased, while the radiation-induced cachexia was mitigated. TNuF downregulated EGFR and VEGF signaling pathways, inhibiting angiogenesis and metastasis.

Specific Comments:

This is technically well performed study but the authors need to address several missing links before it can be considered for publication. Specific points that the authors need to address are as follows:

The molecular mechanism(s) by which TNuF exhibits its effects are not clear? For example, whether deletion of EGFR/VEGFR by siRNA abrogates the observed effects      of TNuF should be analyzed?

Acute toxicity studies should be performed to establish the safety of TNuF.

The anti-invasive/anti-migratory effects of TNuF should be analyzed.

Few typographical errors were noted throughout the manuscript and should be corrected.

Author Response

Comment 1: The molecular mechanism(s) by which TNuF exhibits its effects are not clear? For example, whether deletion of EGFR/VEGFR by siRNA abrogates the observed effects of TNuF should be analyzed?

Reply: We thank the reviewer for this valuable comment. Elucidating molecular mechanism of TNuF should certainly make things clearer. In this study, we first showed that the expressions of EGFR, VEGF, HIF-1α and CD31 in tumor tissues were significantly decreased when TNuF was provided. This result suggests that the TNuF treatment is closely associated with inhibition of EGFR and its downstream modulation on HIF-1α/VEGF/CD31 (Figure 4). Our immunofluorescence analysis showed that the fluorescence intensities of EGFR, VEGF, CD31 and HIF-1α in lungs of mice treated with TNuF were also decreased, indicating that TNuF plays a significant role in regulation of EGFR and its downstream modulation on HIF-1α/VEGF/CD31 in lung metastasis (Figure 5). We further discussed in pages 20-21 that there are strong connections between the active components and anti-invasive/anti-migratory effects of TNuF. Using deletion of EGFR/VEGFR by siRNA to abrogate the observed effects of TNuF is certainly a good strategy, while it is somewhat beyond the scope of the present study and will be explored in the future study.

Comment 2: Acute toxicity studies should be performed to establish the safety of TNuF.

Reply: We appreciate the reviewer’s concern. The fact is that the safety of TNuF has been established, which generally meets the standards of safety and quality of FDA. Several papers have reported that TNuF improved body weight maintenance and serum albumin levels in clinical head and neck cancer patients [1], conforming the safety and beneficial effects of TNuF. On the basis of the components listed on the trial total nutritional formula (Nutrawell) (Table 1), TNuF is a protein- and energy-dense oral nutritional supplement in addition to vitamins, selenium, and CoQ10, that are generally recognized as safe (GRAS). There have been no reports with respect to any adverse side effects of oral level of coenzyme Q10, vitamins and other micronutrients [2-4]. The present data indeed demonstrate that TNuF at a dose of 1 g·day−1 improves cancer cachexia, radiation-induced tumor death, mitigation of radiation-induced cachexia and inhibition of tumor cell distant metastasis, doubling the importance in nutritional management for lung cancer. In terms of the dosage of each ingredient, it is actually far below the value as a possible toxic hazard set by FDA.

Yeh KY, Wang HM, Chang JW, Huang JS, Lai CH, Lan YJ, Wu TH, Chang PH, Wang H, Wu CJ, et al. Omega-3 fatty acid-, micronutrient-, and probiotic-enriched nutrition helps body weightstabilization in head and neck cancer cachexia. Oral Surg Oral Med Oral Pathol Oral Radiol 2013;116:41-8. Ferrante KL, Shefner J, Zhang H, Betensky R, O'Brien M, Yu H, Fantasia M, Taft J, Beal MF, Traynor B, et al. Tolerance of high-dose (3,000 mg/day) coenzyme Q10 in ALS. Neurology 2005;65:1834-6. Xing H, Zheng S, Zhang Z, Zhu F, Xue H, Xu S. Pharmacokinetics of Seleniumin Healthy Piglets After Different Routes of Administration: Application of Pharmacokinetic Data to the Risk Assessment of Selenium. Biol Trace Elem Res 2019; doi: 10.1007/s12011-019-1644-7. Institute of medicine, food and nutrition board. Dietary reference intakes: vitamin C, vitamin E, selenium, and carotenoids. National academy press, Washington, DC, 2000.

Comment 3: The anti-invasive/anti-migratory effects of TNuF should be analyzed.

Reply: We thank the reviewer’s suggestion. The anti-invasive/anti-migratory effects of TNuF was firstly demonstrated in Figure 5A-D. We subsequently discussed that TNuF is able to suppress metastasis of tumor cells and expression of tumor-induced VEGF, EGFR, HIF-1α and CD31 in lung tissues (Page 20 to 21). The overall outcome is that TNuF constrains cancer cells from invasion and metastasis by suppressing epithelial cell proliferation and blood vessel formation. By taking the reviewer’s suggestion, we further discuss the possible active component, selenium, and the anti-metastasis mechanism of TNuF in the discussion section (Page 21).

Comment 4: Few typographical errors were noted throughout the manuscript and should be corrected.
Reply: We thank the reviewer’s suggestion for this concern. The typographical errors have been amended and the language has been considerably improved.

Round 2

Reviewer 1 Report

In each figure, when performing statistical analyses, stars and the corresponding p value need to be checked. Does “two stars” indicate p<0.001 or P<0.01? How about “three stars”?

Author Response

We thank the reviewer’s suggestion. *P<0.05, **P<0.01, ***P<0.001 when compared to the indicated group. All mistakes have been corrected and every figure has been rechecked in the revised manuscript.

Reviewer 2 Report

The authors have addressed all my concerns. Thanks.

Author Response

Reply: We thank the reviewer’s suggestion.
